# Discovering Global False Negatives On the Fly for Self-supervised Contrastive Learning

**Vicente Balmaseda** [1]  **Bokun Wang** [1]  **Ching-Long Lin** [2]  **Tianbao Yang** [1]

## Abstract

In self-supervised contrastive learning, negative pairs are typically constructed using an anchor image and a sample drawn from the entire dataset, excluding the anchor. However, this approach can result in the creation of negative pairs with similar semantics, referred to as "false negatives", leading to their embeddings being falsely pushed apart. To address this issue, we introduce GLOFND, an optimization-based approach that automatically learns on the fly the threshold for each anchor data to *identify* its false negatives during training. In contrast to previous methods for false negative discovery, our approach *globally* detects false negatives across the entire dataset rather than locally within the mini-batch. Moreover, its per-iteration computation cost remains independent of the dataset size. Experimental results on image and image-text data demonstrate the effectiveness of the proposed method. Our implementation is available at "https://github.com/vibalcam/GloFND".

## 1. Introduction

Representation learning is a fundamental problem in machine learning that aims to learn a good representation of the data for downstream tasks. Conventional supervised approaches rely on large quantities of high-quality labeled data, which is hard to collect. Recently, self-supervised learning has achieved promising performance for image representation learning (Chen et al., 2020a; Grill et al., 2020). Its success extends to bimodal learning (Radford et al., 2021) and semi-supervised learning (Chen et al., 2020b). These methods exploit unlabeled data to acquire general-purpose representations that exhibit robust performance and transfer-

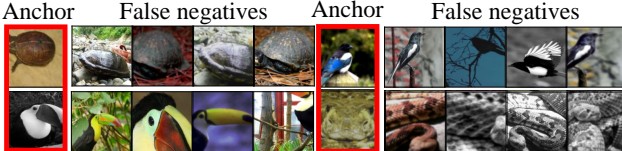

Figure 1: Examples of false negative images seen during training on ImageNet. The left column is the anchor image and the rest are observed false negative samples.

ability across diverse downstream tasks.

Notably, many self-supervised learning approaches center around contrastive learning. Contrastive learning operates on a straightforward principle: it seeks to bring together the embeddings of positive (similar) pairs while simultaneously pushing apart those of negative (dissimilar) pairs. This principle is combined with well-chosen data augmentations to improve the model's invariance to non-semantic variations.

In the absence of reliable labels to determine whether a pair of data is positive or negative, many methods resort to instance discrimination. To this end, positive pairs are defined as distinct augmented views of the anchor data, while negative pairs are generated by sampling from the whole dataset excluding the anchor data, irrespective of their semantics (Chen et al., 2020a; Yuan et al., 2022). However, negative pairs produced through this method lack reliability. Specifically, augmented views from images sharing similar semantic meanings are incorrectly deemed negative, leading to their embeddings being pushed apart. This inadvertently encourages the model to discard crucial semantic information. We term these undesirable negative pairs as false negatives (FN). Figure 1 illustrates examples of false negatives encountered during training on ImageNet, where the anchor images are different from their negative samples yet semantically similar.

The presence of false negatives detrimentally impacts the representations learned through contrastive learning (Saunshi et al., 2019), with this effect becoming more pronounced in large-scale datasets featuring numerous semantic concepts (Chen et al., 2022). For instance, during Sog-CLR (Yuan et al., 2022) pretraining on ImageNet100 (100 classes), approximately 1% of all negative pairs are false

---

[1]Department of Computer Science and Engineering, Texas A&M University, College Station, USA [2]. Correspondence to: Vicente Balmaseda <vibalcam@tamu.edu>.

*Proceedings of the $42^{nd}$ International Conference on Machine Learning*, Vancouver, Canada. PMLR 267, 2025. Copyright 2025 by the author(s).

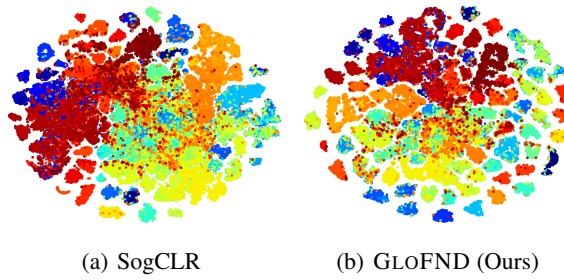

(a) SogCLR      (b) GLoFND (Ours)

Figure 2: ImageNet100 features (t-SNE projected) learned by SogCLR (Yuan et al., 2022) and GLoFND (this work).

negatives. This translates to around 20,000 false negatives per batch with a batch size of 1024, and about 325 with a batch size of 128. The presence of such false negatives during training can significantly degrade the quality of learned representations: a linear classifier trained on top of such representations achieves up to 10% lower accuracy in a semi-supervised setting compared to representations learned without the false negatives.

Given an approach to identify false negatives, we can take corrective actions such as filtering them from the training set (i.e., false negative elimination) or incorporating them as additional positive pairs (i.e., false negative attraction) (Huynh et al., 2022). However, confidently identifying the potential false negatives in the absence of labels poses a challenging problem. The desire to eliminate false negatives is motivated by the goal of improving representation learning, yet the identification of these instances may necessitate some level of semantic knowledge to determine whether two pairs are indeed negative. Looking at Figure 1, we can observe some of the false negatives are not straightforward to identify, especially after data augmentation.

Previous works addressing this problem fall into two categories: local (batch-wise) and global (dataset-wise) approaches. Local methods (Zheng et al., 2021; Huynh et al., 2022) identify false negatives for an anchor by assessing its similarities or adjacency to other data within the same mini-batch. However, the most similar or adjacent item to the anchor in the mini-batch may not necessarily be semantically similar to the anchor in the entire data space, particularly when the mini-batch size is small. The global approach IFND (Chen et al., 2022) aims to discover false negatives for each anchor in the whole dataset. However, their method involves clustering the entire dataset at specific epochs, which could be computationally expensive for large-scale datasets.

This paper addresses existing limitations in false negative *detection* by introducing a novel algorithm, named **Glo**bal **F**alse **N**egative **D**iscovery (GLoFND), which learns global and dynamic thresholds for each anchor in the dataset. This enables the selection of the top-$\alpha\%$ most similar negative

data points from the entire dataset on the fly, with top-$\alpha\%$ being the set above the $(1-\alpha)$-quantile ($\alpha \in [0, 1]$). The GLoFND algorithm alternates between two key steps: i) Updating the per-anchor thresholds by SGD to solve a convex optimization problem of finding a threshold that can filter out the top-$\alpha\%$ of a set of scores. ii) Updating the parameters of the encoder network by using a stochastic gradient estimator of the modified contrastive loss that takes care of the false negatives identified via the learned thresholds (e.g., excluding them).

GLoFND can be integrated with various CL techniques with minimal computational overhead. It effectively identifies false negatives for each sample, offering flexibility in how these are addressed. We demonstrate the empirical success of our method in unimodal, bimodal, and semi-supervised contrastive learning on several CL techniques without using a large batch size. Figure 2(a) and 2(b) qualitatively showcases that identifying and removing false negatives using GLoFND achieves better separation between the learned representations of different classes. One example of this observation is that clusters close to the periphery appear more tightly packed and distinct.

## 2. Related Work

**Self-supervised learning (SSL).** SSL has garnered substantial attention for its capacity to generate general-purpose representations from unlabeled data, facilitating scalability to large-scale datasets (Gui et al., 2023). SSL learns a data encoder network by leveraging *intrinsic* relationships within the data. The encoder network is then used to learn predictive models in downstream tasks through transfer learning. Noteworthy applications span computer vision (Kolesnikov et al., 2019), natural language processing (Lan et al., 2019), and healthcare (Sowrirajan et al., 2021), among others.

Early efforts in SSL formulated pretext tasks to enable models to learn representations from unlabeled data. Examples include predicting the relative offset between two patches within the same image (Doersch et al., 2016), solving jigsaw puzzles (Noroozi & Favaro, 2017), colorizing grayscale images (Zhang et al., 2016), and unsupervised deep clustering (Caron et al., 2019). However, these methods necessitate carefully crafted pretext tasks, which may not always apply to diverse domains, leading to a lack of generality.

**Contrastive learning (CL).** CL has emerged as a prevalent paradigm in SSL, primarily grounded in instance discrimination (Wu et al., 2018; Zhao et al., 2021). This approach employs contrastive losses, compelling the model to bring the embedding vectors of positive pairs closer while simultaneously pushing those of negative pairs apart. In essence, it promotes the learning of representations with high similarity among positive pairs and low similarity among negative

pairs. Positive pairs can be easily generated from different views of the same image. However, generating quality negative pairs is more challenging. MoCo (He et al., 2020) addresses this through (i) a momentum encoder network that generates representations of images for contrast with the anchor, and (ii) a long queue to provide a large number of negative samples. SimCLR (Chen et al., 2020a) instead uses a large batch size and data augmentations, generating negative pairs with augmented views of images other than the anchor image. However, its performance degrades a lot as the batch size decreases. To address this issue, Yuan et al. (2022) propose a stochastic algorithm called SogCLR that does not rely on large batch size. Ge et al. (2024) generates negatives that preserve superfluous instead of semantic features. Shah et al. (2022) introduced a max-margin criterion inspired by support vector machines (SVMs). While these methods have shown promising results, they overlook the semantic relationship when generating negative pairs. Despite two images being semantically similar, their augmented views are treated as negative pairs, a phenomenon referred to as "false negatives". The false negatives result in the loss of crucial semantic information, consequently impacting representation learning (Saunshi et al., 2019).

**Semantic-aware CL.** Recently, several studies have enhanced instance-discrimination-based CL by leveraging the underlying semantics. Qiu et al. (2023) introduce the iSogCLR algorithm that learns individualized temperatures for each sample depending on the frequency of its semantics to increase the tolerance of false negatives. In contrast, GLOFND tackles the more challenging task of *detecting* each sample's false negatives, thus providing more freedom on how to deal with them. Supervised CL (SupCon) (Khosla et al., 2021) has demonstrated that employing the CL objective with labels to define positive and negative pairs (i.e., avoiding false negatives) can be more effective than the conventional supervised cross-entropy loss. Weakly supervised CL (WCL) (Zheng et al., 2021) constructs an undirected graph based on auxiliary embeddings of mini-batch data, whose connected components are used to define weak labels for the SupCon. Huynh et al. (2022) adopt a different approach called FNC, addressing false negatives for each anchor by selecting the top $k$ similar samples in the batch. The limitation of the last two works is that the top similar negative samples in the batch may not be reliable false negatives when the mini-batch size is small.

Instead of detecting false negatives within the mini-batch, Chen et al. (2022) introduce an incremental dataset-wide clustering-based approach. At specific epochs, embeddings are computed for all samples in the dataset, followed by clustering using $k$-means. Pairs of samples within the same cluster are designated as false negatives. Nevertheless, this approach entails high computational costs, particularly for large-scale datasets, due to the necessity of computing embeddings for the entire dataset and subsequently applying $k$-means clustering. Hence, there remains a need for a global (dataset-wise) false-negative discovery approach that is agnostic to batch size and scalable for large-scale datasets.

## 3. CL with Global False Negative Identification

Let $\mathcal{D} = \{\mathbf{x}_1, \ldots, \mathbf{x}_n\}$ denote a dataset of size $n$, and let $\mathcal{P}$ be a collection of data augmentation operators. $E_{\mathbf{w}}(\cdot)$ represents an encoder network parametrized by $\mathbf{w} \in \mathbb{R}^d$.

The global contrastive objective (Yuan et al., 2022) $\mathcal{L}_{\mathrm{GCL}}(\mathbf{w})$ contrasts each $\mathbf{x}_i \in \mathcal{D}$ with negative data $\mathcal{S}_i^- = \{A(\mathbf{x}) \mid \forall A \in \mathcal{P}, \forall \mathbf{x} \in \mathcal{D} \backslash \{\mathbf{x}_i\}\}$ in the whole dataset. Let $\mathbf{z}_i = E_{\mathbf{w}}(A(\mathbf{x}_i))$, $\mathbf{z}_i' = E_{\mathbf{w}}(A'(\mathbf{x}_i))$ denote the embeddings and $\mathrm{sim}(\cdot, \cdot)$ the cosine similarity. Then,

$$\mathcal{L}_{\mathrm{GCL}}(\mathbf{w}) = \mathbf{E}_{\mathbf{x}_i \in \mathcal{D}, A, A' \in \mathcal{P}} \left[ \ell(\mathbf{w}; \mathbf{x}_i, A, A') \right], \qquad (1)$$

$$\ell(\mathbf{w}; \mathbf{x}_i, A, A') = -\tau \log \frac{\exp\left(\mathrm{sim}(\mathbf{z}_i, \mathbf{z}_i')/\tau\right)}{\sum_{\mathbf{x} \in \mathcal{S}_i^-} \exp(\mathrm{sim}(\mathbf{z}_i, E_{\mathbf{w}}(\mathbf{x}))/\tau)},$$

While GLOFND is not restricted to any CL technique, the following sections present how GLOFND can be integrated with the global contrastive loss. GLOFND's application to other CL techniques can be done in a similar manner and we empirically show its effectiveness in Section 4. Since GLOFND's focus is on *detecting* false negatives, it does not restrict what actions to take with the detected false negatives. This paper will consider the straightforward approach of filtering them from the loss, leaving how to make the best use of false negatives for future work.

### 3.1. Learning Dynamic Per-Anchor Thresholds

The challenge in the false negative discovery problem appears akin to a chicken-and-egg dilemma: the reliable identification of false negatives demands good representations, yet achieving quality representation learning necessitates detecting and dealing with false negatives. Thus, our approach starts with a *sufficiently* pre-trained encoder network $E_{\mathbf{w}}$ (this can be done with a warm-up stage using existing CL methods) and further refines it by systematically and dynamically eliminating the identified false negatives. Moreover, we assume that the top $\alpha\%$ most similar negative data share similar semantics with the anchor data based on their current representations, where $\alpha$ is a hyper-parameter to be tuned that allows adapting to different settings. We will identify as false negatives of the anchor $\mathbf{x}_i$ the negative data with similarity scores above the $(1 - \alpha)$-quantile ($\alpha \in [0, 1]$).

The hyperparameter $\alpha$ allows GLOFND to adapt to different definitions of false negatives, which are inherently dependent on the desired level of granularity. For example, in a dataset like ImageNet, consider two classification settings: (1) a coarse-grained task of classifying between cars and

animals, and (2) fine-grained task of classifying dog breeds. Two images of a dog from different breeds might be considered a false negative in the coarse-grained case (1), but not in the fine-grained one (2). Consequently, the optimal percentage of false negatives, controlled by $\alpha$, varies with the chosen granularity. By adjusting $\alpha$, GLOFND can flexibly align with different levels of semantic resolution, i.e., granularity. Its value can be set based on prior knowledge (e.g., expected rate of false negatives or desired granularity of learned representations) or tuned like other hyperparameters, such as the temperature in contrastive learning.

Previous work (Huynh et al., 2022) either selects the top-$k$ most similar negatives or sets a threshold on similarity scores. However, the former involves the expensive computation and ranking of cosine similarities across the entire dataset, while the latter presents challenges due to the need for manually crafted scheduling of the threshold for each anchor. Moreover, both approaches are problematic as similarity scores change when we update the parameters of the encoding network.

Instead, we choose an optimization-based approach to automatically *learn* the per-anchor threshold $\lambda_i$ to select the top $\alpha\%$ most similar negative data for the $i$-th anchor. To achieve this, we cast the problem of finding the $(1 - \alpha)$-quantile of all similarity scores between $\mathbf{x}_i$ and all other samples, i.e., $R_i = \{\text{sim}(E_\mathbf{w}(\mathbf{x}_i), E_\mathbf{w}(\mathbf{x})) \mid \mathbf{x} \in \mathcal{S}_i^-\}$ as the following optimization problem (Ogryczak & Tamir, 2003):

$$\lambda_i = \arg \min_{\boldsymbol{\nu} \in [-1,1]} \nu\alpha + \frac{1}{|R|} \sum_{r \in R_i} (r - \nu)_+ \qquad (2)$$

Then, we will have a set of threshold $\{\lambda_i\}_{i \in [|\mathcal{D}|]}$. The following lemma shows that the solution $\lambda_i$ to (2) can be used to select the top-$\alpha\%$ most similar negative data.

**Lemma 3.1.** *Let $k = \lceil \alpha|\mathcal{S}_i^-|\rceil$. The solution $\lambda_i$ to (2) is either the $k$-th largest value or between $k$-th and $(k + 1)$-th largest value in the set $R_i$.*

We modify the contrastive loss in (1) to eliminate the false negatives, yielding the following optimization problem:

$$\min_\mathbf{w} \mathcal{L}_{\text{GCL}}(\mathbf{w}, \boldsymbol{\lambda}) = \frac{1}{n} \sum_{\mathbf{x}_i \in \mathcal{D}} \mathbf{E}_{A,A'}[\ell(\mathbf{w}, \lambda_i; \mathbf{x}_i)], \qquad (3)$$

$$\ell(\mathbf{w}, \lambda_i; \mathbf{x}_i) = -\text{sim}(\mathbf{z}_i, \mathbf{z}_i') + \tau \log(|\widetilde{\mathcal{S}}_i^-|g(\mathbf{w}, \lambda_i; \mathbf{x}_i, \widetilde{\mathcal{S}}_i^-)),$$

$$g(\mathbf{w}, \lambda_i; \mathbf{x}_i, \widetilde{\mathcal{S}}_i^-) = \frac{1}{|\widetilde{\mathcal{S}}_i^-|} \sum_{\mathbf{x} \in \widetilde{\mathcal{S}}_i^-} \exp(\text{sim}(\mathbf{z}_i, E_\mathbf{w}(\mathbf{x}))/\tau),$$

where $\widetilde{\mathcal{S}}_i^- = \{\mathbf{x} \mid \mathbf{x} \in \mathcal{S}_i^-, \text{sim}(\mathbf{z}_i, E_\mathbf{w}(\mathbf{x})) \leq \lambda_i\}$ is obtained by removing the false negatives (identified via the threshold $\lambda_i$) in the negative dataset $\mathcal{S}_i^-$ for anchor $\mathbf{x}_i$.

Note that $\min_\mathbf{w} \mathcal{L}_{\text{GCL}}(\mathbf{w}, \boldsymbol{\lambda})$ can be viewed as a stochastic bilevel optimization problem (Ghadimi & Wang, 2018)

since the minimization of $\mathcal{L}_{\text{GCL}}(\mathbf{w}, \boldsymbol{\lambda})$ involves the solution $\boldsymbol{\lambda}$ to a lower level problem in (2). However, the problem in (3) is more challenging that most bilevel problems in the literature (Ghadimi & Wang, 2018; Ji et al., 2021) because the lower-level problem in (2) is non-smooth and non-strongly convex while the upper-level function $\mathcal{L}_{\text{GCL}}(\mathbf{w}, \boldsymbol{\lambda})$ is non-differentiable to $\boldsymbol{\lambda}$. To tackle this challenge, we just ignore the hypergradient of $\boldsymbol{\lambda}$ in terms of $\mathbf{w}$, which has been used in model-agnostic meta-learning (Finn et al., 2017).

### 3.2. GLOFND for Unimodal CL

We propose an efficient algorithm called GLOFND for dynamically discovering and eliminating the false negatives in contrastive learning. GLOFND can be combined with previous contrastive learning algorithms, e.g., SogCLR (Yuan et al., 2022). In each iteration, SogCLR + GLOFND first randomly samples a batch of data $\mathcal{B} \subset \mathcal{D}$ and data augmentations $A, A'$. Then, it alternatively executes two steps: (i) updating the thresholds $\lambda_i, i \in \mathcal{B}$; (ii) removing the identified false negatives from the loss function and updating the parameters $\mathbf{w}$ of the encoding network. Notably, step (i), GLOFND 's computation of $\lambda_i$, is independent of the specific contrastive loss used, as it relies solely on the embedding similarity of negative pairs. The contrastive loss only influences how the filtered false negatives are handled during training.

#### 3.2.1. UPDATING THE THRESHOLD $\lambda$

First, the threshold $\lambda_i$ can be updated by calculating the stochastic subgradient of (2) and employing the regular SGD update. Given the predetermined sampled negative data of $\mathbf{x}_i$ in the mini-batch, i.e., $\mathcal{B}_i^- = \{A(\mathbf{x}), A'(\mathbf{x}) \mid \mathbf{x} \in \mathcal{B}\backslash\{\mathbf{x}_i\}\} \subset \mathcal{S}_i^-$, we can compute an stochastic estimator $\widehat{\nabla}_{\lambda_i}$ of the subgradient of (2) w.r.t. $\lambda_i$. Then, we update those $\lambda_i$'s that correspond to those sampled anchors $\mathbf{x}_i \in \mathcal{B}$ while keeping others unchanged, i.e.,

$$\widehat{\nabla}_{\lambda_i} = \alpha - \frac{1}{|\mathcal{B}_i^-|} \sum_{\mathbf{x} \in \mathcal{B}_i^-} \mathbb{I}(\text{sim}(\mathbf{z}_i, E_\mathbf{w}(\mathbf{x})) > \lambda_i)$$

$$\lambda_i \leftarrow \begin{cases} \Pi_{[-1,1]}\left[\lambda_i - \theta\widehat{\nabla}_{\lambda_i}\right], & \mathbf{x}_i \in \mathcal{B}, \\ \lambda_i, & \mathbf{x}_i \notin \mathcal{B}, \end{cases} \qquad (4)$$

where $\mathbb{I}(\cdot)$ is the indicator function, $\Pi_{[-1,1]}[\cdot]$ denotes the projection onto the interval $[-1, 1]$ due to that are similarity scores are in $[-1, 1]$, and $\theta$ is the learning rate of $\lambda_i$. In this way, we keep track of a threshold $\lambda_i$ for detecting global false negatives across the whole dataset for each anchor $\mathbf{x}_i \in \mathcal{D}$, while ensuring that computation remains minibatch-wise.

### 3.2.2. Updating Encoder Network $E_{\mathbf{w}}$

For each anchor $\mathbf{x}_i \in \mathcal{B}$, we eliminate the false negatives identified through the threshold $\lambda_i$ from its negative data batch $\mathcal{B}_i^-$, resulting in $\widetilde{\mathcal{B}}_i^- = \{\mathbf{x} \mid \mathbf{x} \in \mathcal{B}_i^-, \text{sim}(\mathbf{z}_i, E_{\mathbf{w}}(\mathbf{x})) \leq \lambda_i\}$. Following the SogCLR algorithm (Yuan et al., 2022), we use a moving average estimator of $g(\mathbf{w}, \lambda_i; \mathbf{x}_i, \mathcal{S}_i^-)$ for each anchor $\mathbf{x}_i \in \mathcal{D}$ to alleviate the requirement of a large batch size. For each $\mathbf{x}_i$, we maintain a scalar $u_i$ to estimate $g(\mathbf{w}, \lambda_i; \mathbf{x}_i, \mathcal{S}_i^-)$ as

$$u_i \leftarrow \begin{cases} (1-\gamma)u_i + \gamma\widehat{g}(\mathbf{w}, \lambda_i; \mathbf{x}_i, \widetilde{\mathcal{B}}_i^-), & \mathbf{x}_i \in \mathcal{B}, \\ u_i, & \mathbf{x}_i \notin \mathcal{B}, \end{cases} \quad (5)$$

$$\widehat{g}(\mathbf{w}, \lambda_i; \mathbf{x}_i, \widetilde{\mathcal{B}}_i^-) = \frac{1}{|\widetilde{\mathcal{B}}_i^-|} \sum_{\mathbf{x} \in \widetilde{\mathcal{B}}_i^-} \exp(\text{sim}(\mathbf{z}_i, E_{\mathbf{w}}(\mathbf{x}))/\tau),$$

where $\gamma \in [0,1]$ is the parameter and $\widehat{g}(\mathbf{w}, \lambda_i; \mathbf{x}_i, \widetilde{\mathcal{B}}_i^-)$ is an stochastic estimator of $g(\mathbf{w}, \lambda_i; \mathbf{x}_i, \widetilde{\mathcal{S}}_i^-)$. Finally, we can update $\mathbf{w}$ by computing a stochastic estimator of the gradient of $\mathcal{L}_{\text{GCL}}(\mathbf{w}, \boldsymbol{\lambda})$ in (3) w.r.t. the parameters of encoding network $\mathbf{w}$ as:

$$\widehat{\nabla}_{\mathbf{w}} = \frac{1}{|\mathcal{B}|} \sum_{\mathbf{x}_i \in \mathcal{B}} -\nabla_{\mathbf{w}}\text{sim}(\mathbf{z}_i, \mathbf{z}_i') + \frac{\tau \nabla_{\mathbf{w}}\widehat{g}(\mathbf{w}, \lambda_i; \mathbf{x}_i, \widetilde{\mathcal{B}}_i^-)}{u_i}.$$

The whole algorithm, incorporating GLoFND to address the false negative issue in SogCLR through filtering, is outlined in Algorithm 1. Noteworthy differences compared to the vanilla SogCLR algorithm are highlighted in blue. Note that GLoFND adds little overhead, since it just requires basic matrix computations and runs in $O(B^2)$, which is relatively negligible compared to the cosine similarity and forward/backward computations.

---

**Algorithm 1** SogCLR + GLoFND

---

1: **Initialize:** $\mathbf{w} \in \mathbb{R}^d$, initialize $\mathbf{u} \in \mathbb{R}^n$ and $\boldsymbol{\lambda} \in \mathbb{R}^n$
2: **for** $t = 1, \ldots, T$ **do**
3:     Draw a batch of $B$ samples $\mathcal{B} \subset \mathcal{D}$ and data augmentations $A, A'$, and construct $\mathcal{B}_i^- = \{A(\mathbf{x}), A'(\mathbf{x}) \mid \mathbf{x} \in \mathcal{B}\backslash\{\mathbf{x}_i\}\}$ for each $\mathbf{x}_i \in \mathcal{B}$
4:     **for** $\mathbf{x}_i \in \mathcal{B}$ **do**
5:         Update $\lambda_i$ according to (4)
6:         Construct $\widetilde{\mathcal{B}}_i^-$ by excluding the false negatives identified via $\lambda_i$ and compute $\widehat{g}(\mathbf{w}; \mathbf{x}_i, A, \widetilde{\mathcal{B}}_i^-)$
7:         Update $u_{i,t}$ according to (5)
8:     **end for**
9:     Compute the gradient estimator $\widehat{\nabla}_{\mathbf{w}}$
10:    Update $\mathbf{w}$ by the momentum or Adam method
11: **end for**

---

### 3.3. Extension to Bimodal CL

Our approach GLoFND can be extended to resolve the global false negative discovery in bimodal CL, e.g.,

CLIP (Radford et al., 2021). This can be achieved by learning a threshold for each instance for two modalities and following the same general procedure as for unimodal CL. In this section, we provide a brief overview of GLoFND's extension to bimodal CL.

Let $\mathcal{D} = \{(\mathbf{x}_1, \mathbf{t}_1), \ldots, (\mathbf{x}_n, \mathbf{t}_n)\}$ be a set of image-text pairs, and denote the encoder network by $E_I$ for images and the encoder network by $E_T$ for text parametrized by $\mathbf{w}$. For each $(\mathbf{x}_i, \mathbf{t}_i) \in \mathcal{D}$, the negative dataset for each anchor image $\mathbf{x}_i$ is $\mathcal{S}_{I,i}^- = \{\mathbf{t} \mid \forall(\mathbf{x}, \mathbf{t}) \in \mathcal{D} \setminus \{\mathbf{x}_i, \mathbf{t}_i\}\}$ while the negative dataset for each anchor text $\mathbf{t}_i$ is $\mathcal{S}_{T,i}^- = \{\mathbf{x} \mid \forall(\mathbf{x}, \mathbf{t}) \in \mathcal{D} \setminus \{\mathbf{x}_i, \mathbf{t}_i\}\}$. Let $\mathbf{z}_{I,i}$ be the representation of the $i$-th anchor image $\mathbf{x}_i$ and $\mathbf{z}_{T,i}$ be the representation of the $i$-th anchor text $\mathbf{t}_i$, and $\lambda_{I,i}, \lambda_{T,i} \in [-1, 1]$ their respective associated thresholds defined through (2). Then, the problem is formulated as:

$$\min_{\mathbf{w}} \frac{1}{n} \sum_{(\mathbf{x}_i, \mathbf{t}_i) \in \mathcal{D}} \mathbf{E}[\ell(\mathbf{w}, \lambda_{I,i}, \lambda_{T,i}; \mathbf{x}_i, \mathbf{t}_i)]$$

$$\ell(\mathbf{w}, \lambda_{I,i}, \lambda_{T,i}; \mathbf{x}_i, \mathbf{t}_i) = -2\text{sim}(\mathbf{z}_{I,i}, \mathbf{z}_{T,i})$$
$$+ \tau \log|\widetilde{\mathcal{S}}_{I,i}^-|g_I(\mathbf{w}, \lambda_{I,i}; \mathbf{x}_i, \widetilde{\mathcal{S}}_{I,i}^-)$$
$$+ \tau \log|\widetilde{\mathcal{S}}_{T,i}^-|g_T(\mathbf{w}, \lambda_{T,i}; \mathbf{t}_i, \widetilde{\mathcal{S}}_{T,i}^-),$$

$$g_I(\mathbf{w}, \lambda_{I,i}; \mathbf{x}_i, \widetilde{\mathcal{S}}_{I,i}^-) = \frac{1}{|\widetilde{\mathcal{S}}_{I,i}^-|} \sum_{\mathbf{t} \in \widetilde{\mathcal{S}}_{I,i}^-} \exp(\text{sim}(\mathbf{z}_{I,i}, E_T(\mathbf{t}))/\tau),$$

$$g_T(\mathbf{w}, \lambda_{T,i}; \mathbf{t}_i, \widetilde{\mathcal{S}}_{T,i}^-) = \frac{1}{|\widetilde{\mathcal{S}}_{T,i}^-|} \sum_{\mathbf{x} \in \widetilde{\mathcal{S}}_{T,i}^-} \exp(\text{sim}(E_I(\mathbf{x}), \mathbf{z}_{T,i})/\tau),$$

where $\widetilde{\mathcal{S}}_{I,i}^- = \{\mathbf{t} \mid \mathbf{t} \in \mathcal{S}_{I,i}^-, \text{sim}(\mathbf{z}_{I,i}, E_T(\mathbf{t})) \leq \lambda_{I,i}\}$ and $\widetilde{\mathcal{S}}_{T,i}^- = \{\mathbf{x} \mid \mathbf{x} \in \mathcal{S}_{T,i}^-, \text{sim}(E_I(\mathbf{x}), \mathbf{z}_{T,i}) \leq \lambda_{T,i}\}$.

Then, we can easily extend the GLoFND to the bimodal setting similarly as in the unimodal setting. We refer readers to the Appendix B for more details.

## 4. Experiments

In this section, we evaluate GLoFND in unimodal, semi-supervised unimodal, and bimodal scenarios. It is not our focus to leverage multiple techniques for achieving state-of-the-art performance, but to showcase GLoFND's improvements in *identifying* false negatives across different settings while being scalable to large-scale datasets (with negligible overhead) and compatible with small batch sizes. Additionally, we perform an ablation study to analyze the effect of the different components of GLoFND. We report the score average and standard deviation in parenthesis over 3 runs with different random seeds.

The unimodal experiments are run on a single NVIDIA A30 with 24GB memory size, while the bimodal experiments make use of a multi-node setup with 2 nodes, each with 2

Table 1: Linear evaluation results in unimodal semi-supervised scenario. We train the linear classifiers with different percentages of randomly sampled labeled training data and present their top-1 accuracies (%) on the validation set. We include the overall average and, in parentheses, its improvement WRT SogCLR baseline. We also report the average of recall, precision, and f1-score of the identified false negatives over the final epoch of pretraining.

| Method | Top-1 Accuracy | | | | | False Negatives Identification | | |
| | 100.0% | 10.0% | 1.0% | 0.1% | Average | Precision | Recall | F1-Score |
|---|---|---|---|---|---|---|---|---|
| SogCLR | 76.55 (0.09) | 72.24 (0.14) | 62.92 (0.34) | 34.94 (0.60) | 61.66 | — | — | — |
| + FNC | 77.12 (0.14) | 72.89 (0.25) | 64.29 (0.34) | 36.11 (0.70) | 62.60 (+0.94) | 27.57 (0.03) | 53.67 (0.27) | 36.42 (0.07) |
| + GLOFND | **77.59** (0.03) | **73.36** (0.15) | **65.09** (0.49) | **37.38** (0.97) | **63.36** (+1.70) | **48.40** (0.65) | **58.81** (0.60) | **53.10** (0.31) |

Table 2: Unimodal transfer learning results. We report the overall average and its improvement WRT SogCLR baseline.

| Method | CIFAR10 | CIFAR100 | Food101 | Caltech101 | Cars | DTD | Pets | Flowers | Average |
|---|---|---|---|---|---|---|---|---|---|
| SogCLR | 82.46 (0.36) | 60.2 (0.22) | 59.66 (0.2) | 77.73 (0.17) | 25.99 (0.64) | 57.80 (0.30) | 60.63 (0.34) | 76.91 (0.33) | 62.67 |
| + FNC | 82.77 (0.36) | 60.96 (0.31) | 59.82 (0.14) | 78.34 (0.92) | 26.28 (0.50) | 58.60 (0.29) | 61.67 (0.66) | 78.77 (0.51) | 63.40 (+0.73) |
| + GLOFND | **82.81** (0.23) | **61.94** (0.24) | **59.87** (0.16) | **79.18** (0.52) | **27.88** (0.44) | **58.97 (0.96)** | **63.91** (0.22) | **79.89** (0.09) | **64.31** (+1.64) |

NVIDIA A100 GPUs with 40GB each.

For unimodal and semi-supervised experiments, we use **SogCLR** (Yuan et al., 2022) and compare with FNC (Huynh et al., 2022). FNC computes the top $k$ for negative data within a mini-batch by utilizing a support set. The support set includes additional views for each image, and the similarity scores are averaged across these views. For a fair comparison, we set $k = \alpha|\mathcal{D}|$ and use a support set of size 1. For bimodal experiments, we compare GLOFND with **SogCLR** and **FastCLIP** (Wei et al., 2024).

### 4.1. Unimodal and Semi-supervised Experiments

**Dataset.** We run our experiments on ImageNet100 (Wu et al., 2019), a subset of ImageNet with 100 randomly selected classes (about 128k images), and report scores on its official validation split. Additionally, we examine the transfer learning performance on Food-101 (Bossard et al., 2014), CIFAR-10 and CIFAR-100 (Krizhevsky, 2009), Stanford Cars (Krause et al., 2013), Describable Textures Dataset (DTD) (Cimpoi et al., 2014), Oxford-IIIT Pets (Parkhi et al., 2012), Caltech-101 (Li et al., 2022), and Oxford 102 Flowers (Nilsback & Zisserman, 2008).

**Experiment Setup.** Following previous work (Yuan et al., 2022), we pretrain ResNet-50 (He et al., 2015) with a 2-layer $128 \times 128$ projection head on top of the backbone encoder. We pretrain for 200 epochs with a batch size of 128 and the same set of augmentations as in SogCLR. We use LARS optimizer (You et al., 2017) with square root learning rate scaling ($0.075 \times sqrt(BatchSize)$) and cosine decay schedule without restart. For SogCLR, we set the temperature ($\tau$) to 0.1 and $\gamma = 0.9$. We start using GLOFND when we reach 70 epochs. We use $\alpha = 0.01$, initialize $\lambda_i = 1$, and learn it with Adam with a learning rate of 0.05 ($\beta_1 = 0.9, \beta_2 = 0.98$) during the remaining

epochs. For FNC, we set $\alpha = 0.01$ and tune the starting epoch in $\{10, 30, 50, 70, 90, 110, 130\}$, choosing the value that achieves the best semi-supervised average performance. More details on hyperparameters can be found in Appendix D.1.

**Evaluation.** We evaluate our model in three ways: false negative identification, semi-supervised linear evaluation, and transfer learning. First, given that GLOFND's main objective is false negative identification, we assess its effectiveness to correctly detect false negatives. To construct the ground truth, we compare the labels of each sample pair, if both samples have the same label they are considered a false negative. We report precision, recall, and F1-scores for the final epoch of pretraining. Second, we evaluate GLOFND's ability to achieve better representations through linear evaluation. That is, we freeze the weights of the encoder at the last iteration of pretraining, remove its projection head, and train a linear classifier on top of the encoder's output. We follow a semi-supervised learning setup, where we use different fractions of labeled training data during linear evaluation, i.e., we train on random subsets of 100% (full dataset), 10%, 1%, and 0.1% of the training data. We report each top-1 accuracy on the validation set and average the performance across percentages obtaining the overall semi-supervised score. Lastly, we evaluate the transfer learning performance of the learned representations. We train an $\ell_2$-regularized logistic regression classifier on features extracted from the frozen pretrained network after removing the projector head. For each method, we report linear evaluation and transfer learning results for the model that achieves the highest semi-supervised average performance.

**Results.** We report false negative identification performance and top-1 accuracies by linear evaluation in Table 1. GLOFND achieves significant improvements in false negative identification over FNC, with a 20.83% and 5.14% in-

Table 3: Results in bimodal zero-shot downstream tasks. Datacomp provides the average across 38 tasks, Retrieval averages the performance on 3 image-text retrieval datasets, and IN & Variants averages 7 ImageNet datasets.

| Method | Datacomp | Retrieval | IN & Variants |
|---|---|---|---|
| SogCLR (Wei et al., 2024) | 24.87 (0.13) | 29.28 (0.30) | 18.86 (0.09) |
| + FNC | 24.55 (0.20) | 29.69 (0.19) | 18.69 (0.62) |
| + GLoFND | **25.37** (0.16) | **29.92** (0.35) | **19.44** (0.16) |
| FastCLIP (Wei et al., 2024) | 24.76 (0.26) | **30.36** (0.18) | 19.08 (0.16) |
| + FNC | 24.63 (0.72) | 28.87 (0.93) | 18.51 (0.19) |
| + GLoFND | **25.37** (0.13) | 30.22 (0.32) | **19.38** (0.15) |

crease in mean precision and recall, leading to an F1-score of 53.10%, which is 16.68% higher than FNC. Observe that simply removing the false negatives identified by FNC or GLoFND improves both the semi-supervised and transfer learning performance of SogCLR. GLoFND achieves greater improvements in both scenarios, achieving 1.04%-2.44% improvement in the semi-supervised scenario and an average 1.64% improvement in transfer learning, while increasing the per-epoch computation by only 2% (from 427 s to 435 s). More details can be found in Appendix D.3. Note these improvements are achieved by simply removing the false negatives identified by GLoFND from the loss, while a more careful treatment can potentially improve the performance even further.

## 4.2. Bimodal Experiments

**Datasets.** For bimodal learning, we use the Conceptual Captions 3M (CC3M) (Sharma et al., 2018) dataset. We evaluate the performance by leveraging the Datacomp Benchmark (Gadre et al., 2023), which includes 38 zero-shot downstream tasks. We report the average performance, named Datacomp. For each scenario, we select the model with the best Datacomp average and also report its average performance on two subsets of the tasks: zero-shot image classification on ImageNet-1k (Russakovsky et al., 2015) and 6 ImageNet distribution shift datasets (Wang et al., 2019; Recht et al., 2019; Hendrycks et al., 2021b;a; Barbu et al., 2019) (IN & Variants), and zero-shot cross-modal image-text retrieval on Flickr30K (Plummer et al., 2017), MSCOCO (Lin et al., 2015), and WinoGAViL (Bitton et al., 2022).

**Experiment Setup.** Following previous work (Wei et al., 2024), we use a 12-layer transformer (Vaswani et al., 2017) as the text encoder, and ResNet50 as the vision encoder. All experiments are conducted in a multi-node setting with 2 nodes, each with two A100 40GB GPUs. We pretrain for 37 epochs with a global batch size of 1024. For SogCLR, we start using FNC/GLoFND after 15 epochs, setting $\alpha = 5e - 4$, while for FastCLIP we start after 20 epochs with $\alpha = 1e - 3$. For both losses, we initialize $\lambda_i = 1$ and use Adam updates with a learning rate of 0.05. More details on hyperparameters can be found in Appendix D.2.

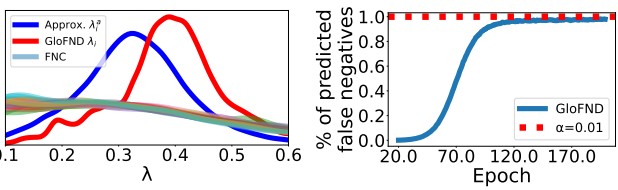

|  (a) Threshold distributions | (b) % of predicted FN |
|---|---|

Figure 3: Analysis of learned $\lambda_i$ for GLoFND with $\alpha = 0.01$. (a) Kernel density estimation of the distributions of GLoFND's $\lambda_i$, the approximated optimal $\lambda_i^a$, and 20 randomly sampled FNC thresholds. (b) Average percentage of negative pairs predicted to be false negatives during training (i.e., $1 - |\widetilde{S}_i^-|/|S_i^-|$).

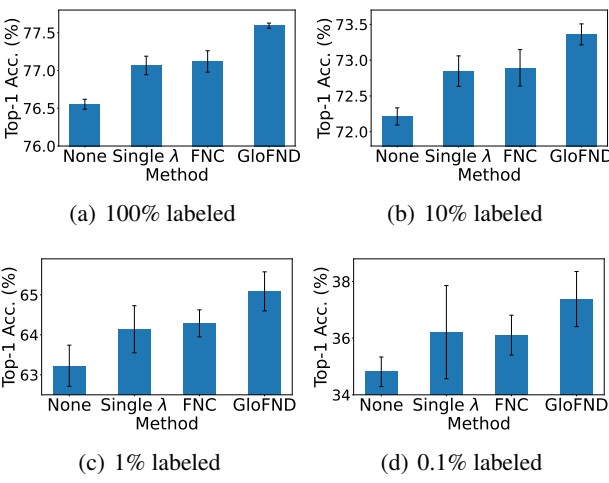

(a) 100% labeled      (b) 10% labeled

(c) 1% labeled      (d) 0.1% labeled

Figure 4: Linear evaluation (top-1 accuracy %) on ImageNet100 for SogCLR without FN identification, with FNC (*top* 1%), with a single learned threshold ($\alpha = 0.1$), and GLoFND ($\alpha = 0.01$).

**Results.** We present the bimodal results in Table 3. Despite using a larger batch compared to the unimodal case, the bimodal scenario proves more challenging for FNC, which generally underperforms compared to the baseline models. In contrast, GLoFND enhances both SogCLR and Fast-CLIP across most metrics. Notably, it improves the overall Datacomp score for both models. These results highlight the benefit of integrating false negative detection into bimodal contrastive losses, demonstrating that GLoFND is an effective approach for this task.

## 4.3. Ablation Study

**Verification of Algorithm Design.** We empirically validate three aspects of GLoFND's design: (i) the necessity to have a global threshold (as opposed to batch-wise), (ii) the necessity to have a different $\lambda_i$ for each anchor $x_i \in \mathcal{D}$, and (iii) the quality of the learned $\lambda_i$ threshold. We will cover (ii) and (iii) in this section and (i) in the next section.

*(ii) Do we need a different threshold for each anchor?*

We first examine the distribution of $\lambda_i$ learned by GLOFND after pretraining. Rather than being concentrated around a single value, we expect it to span a range, indicating that different anchors adopt different thresholds when computing their top $(1 - \alpha)$-quantile. Figure 3(a) illustrates this distribution for ImageNet100 (red line). As expected, $\lambda_i$ varies within the range $[0.1, 0.6]$, highlighting the necessity of a per-anchor threshold.

To empirically validate this, we compare GLOFND, which assigns a distinct $\lambda_i$ per anchor, against a variant that uses a single $\lambda$ for all anchors, referred to as "Single $\lambda$." Figure 4 reports the semi-supervised performance on ImageNet100, demonstrating that GLOFND with per-anchor $\lambda_i$ consistently outperforms the single-threshold variant.

*(iii) How good are the learned $\lambda_i$ thresholds?*

We train SogCLR on ImageNet100 and apply GLOFND ($\alpha = 0.01$) with SGD updates starting at epoch 20. We monitor the percentage of negative pairs predicted to be false negatives, computed as $1 - |\widetilde{\mathcal{S}}_i^-|/|\mathcal{S}_i^-|$, throughout training. We initialize $\lambda_i = 1$ (indicating no false negatives) and expect the percentage to converge to the target $\alpha$. As shown in Figure 3(b), GLOFND successfully reaches and maintains the desired $\alpha$ after a few epochs.

Next, we evaluate the error of the learned $\lambda_i$ relative to its optimal value. Since computing the exact optimal $\lambda_i$ is intractable, requiring similarity calculations for every anchor against all other samples under different augmentations, we approximate it instead. We freeze the network and estimate the optimal threshold $\lambda_i^a$ by randomly selecting 100,000 samples per anchor. Empirically, GLOFND approximates the desired threshold significantly better than FNC with a batch size of 128, achieving a Mean Absolute Error (MAE) of 0.1 and a Root Mean Squared Error (RMSE) of 0.13 ($\lambda_i \in [-1, 1]$), whereas FNC obtains MAE and RMSE of 0.21 and 0.28, respectively. This means GLOFND has less than half the error of FNC. Qualitatively, Figure 3(a) compares the distribution of the learned $\lambda_i$, the approximated $\lambda_i^a$, and the thresholds used by FNC. Since FNC computes thresholds at the mini-batch level, we sample 20 random batches per anchor and plot their respective distributions. The results show that GLOFND learns a $\lambda_i$ distribution that more closely aligns with the desired threshold than FNC.

**Impact of Starting Epoch.** GLOFND requires a *sufficiently* pre-trained network to ensure that the similarity between embeddings reflects semantic similarity. This is achieved by applying GLOFND after a certain number of training epochs. However, the optimal start time involves a trade-off. If GLOFND is applied too early, the embedding space may not be well-formed, leading to incorrect classification of pairs as false negatives. Conversely, if applied too late, the

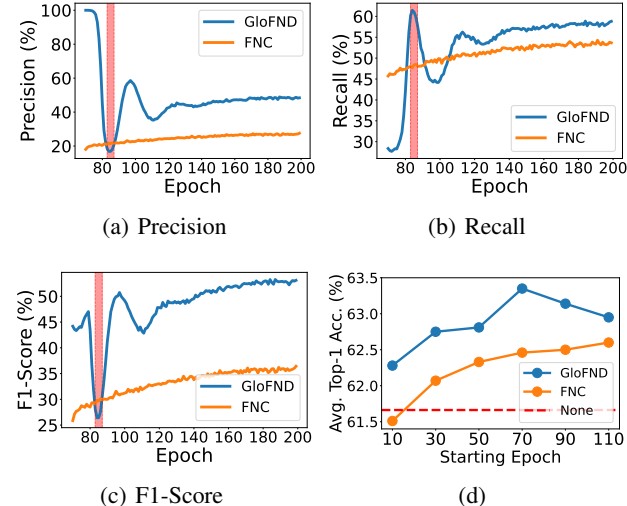

(a) Precision

(b) Recall

(c) F1-Score

(d)

Figure 5: FNC and GLOFND comparison. (a): False negative prediction performance scores for ImageNet100 using the labels as ground truth. We report the per-epoch mean precision, recall, and f1-score for SogCLR + FNC and SogCLR + GLOFND ($\alpha = 0.01$). (b): Starting epoch comparison on linear evaluation performance.

potential benefits of GLOFND may be limited due to insufficient training time. To assess the necessity and impact of this "wait" period, we evaluate GLOFND's semi-supervised performance on ImageNet100 as we vary the start epoch in $\{10, 30, 50, 70, 90, 110\}$, keeping the total number of training epochs fixed at 200. The linear evaluation results, presented in Figure 5(d), show that GLOFND's performance improves as the wait time increases, peaking at 70 epochs. Beyond this point, performance begins to decline. FNC shows a similar trend, with its performance degrading when applied after 110 epochs.

**Computational Efficiency.** When using GloFND with a contrastive method based on embedding similarity (e.g., SimCLR, SogCLR, and CLIP), pairwise similarities are already computed as part of the loss function. Thus, the only additional computations required are: (1) updating the $\lambda_i$ values for the mini-batch samples, which involves a simple gradient computation (Equation 4), and (2) filtering the false negatives, which can be done through simple matrix operations. Both operations involve basic matrix computations and run in linear time with respect to the number of pairs in a batch ($O(B^2)$, where $B$ is the batch size). The computational overhead of GLOFND is minimal compared to the cost of cosine similarity computation and forward/backpropagation. Our experiments on ImageNet100 with a batch size of 128 show that GLOFND introduces only a **2% increase** in per-epoch training time for SogCLR (435.19 s for SogCLR + GLOFND vs. 426.67 s for SogCLR). This overhead is comparable to that of the batch-wise method FNC (434.06 s).

## 4.4. Comparison with Mini-batch Top-$k$ Method

In this section, we assess the necessity of a global threshold by comparing GLOFND's global thresholds to FNC, which computes a threshold for each mini-batch.

**Semi-supervised Linear Evaluation.** We present the semi-supervised linear evaluation performance for different percentages of labeled training data in Figure 4. Both GLOFND and FNC outperform not addressing false negatives, highlighting the importance of handling them. Furthermore, GLOFND consistently outperforms FNC across all settings, with improvements ranging from 0.47% to 1.27%. This demonstrates the advantage of using a global threshold, as opposed to a threshold specific to each mini-batch.

**Quality of Found False Negatives.** We analyze the quality of the false negatives identified by GLOFND and FNC. In Section 4.3 (iii), we discussed how GLOFND matches more closely the optimal dataset-wide threshold than FNC, with half the approximation error. Here, we examine how this improved threshold alignment affects the quality of the false negatives identified. To do so, we calculate the per-epoch mean precision, recall, and F1-score for each method, using the class labels as ground truth (i.e., a pair is considered a false negative if both samples share the same label). As training progresses and the embedding space improves, we expect these metrics to increase, reflecting better alignment between embedding and semantic similarity. Furthermore, for GLOFND, we expect an increase in recall as $\lambda_i$ reaches the desired quantile, capturing more false negatives. This should lead to a decrease in precision due to early representations not being sufficiently pretrained. After some oscillation, GLOFND should follow a steady upward trend.

The results are presented in Figure 5. We observe that GLOFND behaves as expected, with the oscillations diminishing and becoming minimal around epoch 120. Regardless, GLOFND shows a 14.89% average improvement in F1-score, surpassing FNC for all but 4 epochs (indicated by the red area in Figure 5). Moreover, after epoch 120, GLOFND consistently maintains a mean F1-score between 14.64% and 18.05% higher than FNC. This underscores GLOFND's superiority in identifying false negatives.

This is quantitatively illustrated in Figure 6, which shows examples of false negatives identified in a mini-batch by GLOFND and FNC. While the number of false negatives identified by FNC remains constant across mini-batches, GLOFND's dynamic threshold allows this number to vary, adapting to each mini-batch more effectively. For instance, in the second and third rows, FNC's fixed top-$k$ approach results in the selection of negative samples that are not sufficiently similar, leading to errors. In contrast, GLOFND is not constrained to a fixed number and instead selects only the most similar samples according to $\lambda_i$. The opposite

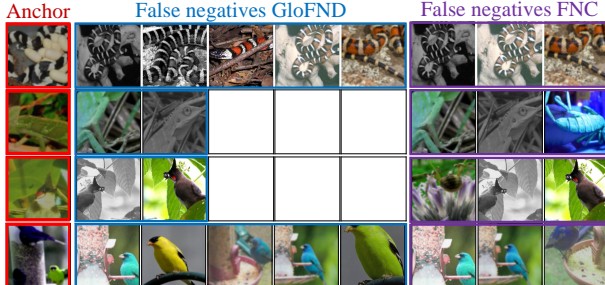

Figure 6: Examples of false negatives identified for ImageNet100 by GLOFND and FNC. The left column shows the anchor images, while the middle and right columns present the false negatives identified by GLOFND and FNC respectively.

occurs in the first and last rows, where GLOFND identifies more false negatives than FNC.

## 5. Conclusions

In this work, we have addressed the problem of *identifying* global false negatives in self-supervised contrastive learning through an optimization-based approach. We propose identifying as false negatives for a given anchor those negative samples whose similarity exceeds the desired quantile across the *entire dataset*. We then introduce GLOFND, an optimization-based method that automatically learns a threshold for each anchor, enabling the identification of its false negatives on the fly. Experimental results demonstrate that GLOFND improves existing contrastive learning methods, both for unimodal and bimodal tasks, with minimal computational overhead. An open question is whether GLOFND could be extended to non-CL methods and whether the parameter $\alpha$ could be individualized.

**Limitations.** Since the focus of this paper is on false negative *detection* for contrastive learning, we address false negatives through filtering. While this straightforward approach has proven effective in our settings, future work could explore more advanced methods that may further enhance downstream performance. Additionally, the benefits of GLOFND and similar false-negative techniques on downstream tasks depend on the proportion of false negatives in the pretraining dataset and how false negatives are defined within the downstream task.

## Acknowledgments

VB, BW and TY were partially supported by National Science Foundation Award #2306572 and #2147253, National Institutes of Health Award #R01HL168116. CL was partially supported by National Institutes of Health Award #R01HL168116.

## Impact Statement

This paper presents work whose goal is to advance the field of Machine Learning. There are many potential societal consequences of our work, none of which we feel must be specifically highlighted here.

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

## A. Proof of Lemma 3.1

*Proof.* For simplicity, we denote that $n_i^- = |S_i^-|$ and $s_j$ is the $j$-th largest value in $\{\text{sim}(\mathbf{z}_i, E_{\mathbf{w}}(\mathbf{x})) \mid \mathbf{x} \in S_i^-\}$, i.e., $s_1 \geq s_2 \geq \ldots \geq s_{n_i^-}$. A subgradient $\phi_i'(\lambda_i)$ of the objective at $\lambda_i$ in (2) is

$$\phi_i'(\lambda_i) = \alpha n_i^- - \sum_{j=1}^{n_i^-} \psi(s_j - \lambda_i), \quad \psi(s_j - \lambda_i) = \begin{cases} 1, & s_j > \lambda_i \\ \epsilon, & s_j = \lambda_i \\ 0, & s_j < \lambda_i, \end{cases}$$

where $\epsilon \in [0, 1]$. We define $k = \lceil \alpha n_i^- \rceil$. $0 \in \partial\phi_i(\lambda_i)$ only happens when $\lambda_i \in [s_{k''}, s_{k'})$ since $k - 1 < \alpha n_i^- \leq k$, where $k' = \max\{j \mid s_j > s_k, j > k\}$, $k'' = \min\{k''', k + 1\}$, $k''' = \max\{j \mid s_j < s_k, j > k\}$. Thus, $s_k$ is a solution to (2). Besides, when $\alpha n_i^-$ is an integer (i.e. $k = \alpha n_i^- = \lceil \alpha n_i^- \rceil$), any value between $[s_k, s_{k+1}]$ is also a solution to (2), which could be different from $s_k$.

$\square$

## B. More Details on Extension to Bimodal CL

Our approach GLOFND can be extended to resolve the global false negative discovery in bimodal CL, e.g., CLIP (Radford et al., 2021). Consider a dataset of image-text pairs $\mathcal{D} = \{(\mathbf{x}_1, \mathbf{t}_1), \ldots, (\mathbf{x}_n, \mathbf{t}_n)\}$, a collection of image augmentation operators $\mathcal{P}_I$, and a collection of text augmentation operators $\mathcal{P}_T$. Suppose that the encoder network $E_I$ for images and the encoder network $E_T$ for text are parametrized by $\mathbf{w}$. For each $(\mathbf{x}_i, \mathbf{t}_i) \in \mathcal{D}$, the negative dataset for each anchor image $\mathbf{x}_i$ is $S_{I,i}^- = \{A_T(\mathbf{t}) \mid \forall A_T \in \mathcal{P}_T, \forall (\mathbf{x}, \mathbf{t}) \in \mathcal{D}\}$ while the negative dataset for each anchor text $\mathbf{t}_i$ is $S_{T,i}^- = \{A_I(\mathbf{x}) \mid \forall A_I \in \mathcal{P}_I, \forall (\mathbf{x}, \mathbf{t}) \in \mathcal{D}\}$. For the $i$-th anchor image $\mathbf{x}_i$ with representation $\mathbf{z}_{I,i}$, the threshold $\lambda_{I,i} \in [-1, 1]$ for finding the top $\alpha\%$ text neighbors among all negatives $S_{I,i}^-$ can be solved by

$$\min_{\nu \in [-1,1]} \phi_{I,i}(\nu),$$

$$\phi_{I,i}(\nu) = \nu\alpha + \frac{1}{|S_{I,i}^-|} \sum_{\mathbf{t} \in S_{I,i}^-} (\text{sim}(\mathbf{z}_{I,i}, E_T(\mathbf{t})) - \nu)_+ .$$

Similarly, we can obtain the threshold $\lambda_{T,i} \in [-1, 1]$ for the $i$-th anchor text $\mathbf{t}_i$. Given the thresholds $\boldsymbol{\lambda}_I, \boldsymbol{\lambda}_T$, the bimodal contrastive loss can be written as

$$\mathcal{L}_{\text{BGCL}}(\mathbf{w}, \boldsymbol{\lambda}_I, \boldsymbol{\lambda}_T) = \frac{1}{n} \sum_{(\mathbf{x}_i, \mathbf{t}_i) \in \mathcal{D}} \mathbf{E}[\ell(\mathbf{w}, \lambda_{I,i}, \lambda_{T,i}; \mathbf{x}_i, \mathbf{t}_i)],$$

$$\ell(\mathbf{w}, \lambda_{I,i}, \lambda_{T,i}; \mathbf{x}_i, \mathbf{t}_i) = -2\text{sim}(\mathbf{z}_{I,i}, \mathbf{z}_{T,i})$$
$$+ \tau \log|S_{I,i}^-| g_I(\mathbf{w}, \lambda_{I,i}; \mathbf{x}_i, \widetilde{S}_{I,i}^-)$$
$$+ \tau \log|S_{T,i}^-| g_T(\mathbf{w}, \lambda_{T,i}; \mathbf{t}_i, \widetilde{S}_{T,i}^-),$$

$$g_I(\mathbf{w}, \lambda_{I,i}; \mathbf{x}_i, \widetilde{S}_{I,i}^-) = \frac{1}{|\widetilde{S}_{I,i}^-|} \sum_{\mathbf{t} \in \widetilde{S}_{I,i}^-} \exp(\text{sim}(\mathbf{z}_{I,i}, E_T(\mathbf{t}))/\tau),$$

$$g_T(\mathbf{w}, \lambda_{T,i}; \mathbf{t}_i, \widetilde{S}_{T,i}^-) = \frac{1}{|\widetilde{S}_{T,i}^-|} \sum_{\mathbf{x} \in \widetilde{S}_{T,i}^-} \exp(\text{sim}(E_I(\mathbf{x}), \mathbf{z}_{T,i})/\tau),$$

where $\widetilde{S}_{I,i}^- = \{\mathbf{t} \mid \mathbf{t} \in S_{I,i}^-, \text{sim}(\mathbf{z}_{I,i}, E_T(\mathbf{t})) \leq \lambda_{I,i}\}$, $\widetilde{S}_{T,i}^- = \{\mathbf{x} \mid \mathbf{x} \in S_{T,i}^-, \text{sim}(E_I(\mathbf{x}), \mathbf{z}_{T,i}) \leq \lambda_{T,i}\}$ are the negative datasets for anchor $(\mathbf{x}_i, \mathbf{t}_i)$ excluding the false negatives identified through the learned thresholds $\lambda_{I,i}, \lambda_{T,i}$.

We extend the GLOFND to the bimodal setting as follows. First, we sample a mini-batch of image-text pairs $\mathcal{B} \subset \mathcal{D}$, sampled image augmentations $A_I$, and text augmentations $A_T$, we construct the sampled negative sets $\mathcal{B}_{I,i}^- = \{A_T(\mathbf{t}) \mid (\mathbf{x}, \mathbf{t}) \in \mathcal{B}\setminus\{(\mathbf{x}_i, \mathbf{t}_i)\}\}$, $\mathcal{B}_{T,i}^- = \{A_I(\mathbf{x}) \mid (\mathbf{x}, \mathbf{t}) \in \mathcal{B}\setminus\{(\mathbf{x}_i, \mathbf{t}_i)\}\}$ for each $(\mathbf{x}_i, \mathbf{t}_i) \in \mathcal{B}$. Given the image embedding $\mathbf{z}_{I,i} = E_I(A_I(\mathbf{x}_i))$ and text embedding $\mathbf{z}_{T,i} = E_T(A_T(\mathbf{t}_i))$ for anchor $(\mathbf{x}_i, \mathbf{t}_i)$, the thresholds $\boldsymbol{\lambda}_I$ for images can be

updated by

$$\widehat{\nabla}_{\lambda_{I,i}} = \alpha - \frac{1}{|\mathcal{B}_{I,i}^-|} \sum_{\mathbf{t} \in \mathcal{B}_{I,i}^-} \mathbb{I}(\text{sim}(\mathbf{z}_{I,i}, E_T(\mathbf{t})) > \lambda_{I,i}),$$

$$\lambda_{I,i} \leftarrow \begin{cases} \Pi_{[-1,1]} \left[ \lambda_{I,i} - \theta \widehat{\nabla}_{\lambda_{I,i}} \right], & (\mathbf{x}_i, \mathbf{t}_i) \in \mathcal{B}, \\ \lambda_{I,i}, & (\mathbf{x}_i, \mathbf{t}_i) \notin \mathcal{B}, \end{cases}$$

Similarly, we can update the thresholds $\boldsymbol{\lambda}_T$ for texts. Given the thresholds $\boldsymbol{\lambda}_I$ for images and thresholds $\boldsymbol{\lambda}_T$ for texts, we can construct $\widetilde{\mathcal{B}}_{I,i}^- = \{\mathbf{t} \mid \mathbf{t} \in \mathcal{B}_{I,i}^-, \text{sim}(\mathbf{z}_{I,i}, E_T(\mathbf{t})) \leq \lambda_{I,i}\}$ and $\widetilde{\mathcal{B}}_{T,i}^- = \{\mathbf{x} \mid \mathbf{x} \in \mathcal{B}_{T,i}^-, \text{sim}(E_I(\mathbf{x}), \mathbf{z}_{T,i}) \leq \lambda_{T,i}\}$ by excluding the false negative images and texts identified via the thresholds $\lambda_{I,i}$ and $\lambda_{T,i}$.

Then, we employ the moving-average estimators $u_{I,i}$, $u_{T,i}$ for $g_I(\mathbf{w}, \lambda_{I,i}; \mathbf{x}_i, \widetilde{\mathcal{S}}_{I,i}^-)$, $g_T(\mathbf{w}, \lambda_{T,i}; \mathbf{t}_i, \widetilde{\mathcal{S}}_{T,i}^-)$, respectively.

$$\widehat{g}_I(\mathbf{w}, \lambda_{I,i}; \mathbf{x}_i, \widetilde{\mathcal{B}}_{I,i}^-) = \frac{1}{|\widetilde{\mathcal{B}}_{I,i}^-|} \sum_{\mathbf{t} \in \widetilde{\mathcal{B}}_{I,i}^-} \exp(\text{sim}(\mathbf{z}_{I,i}, E_T(\mathbf{t}))/\tau),$$

$$\widehat{g}_T(\mathbf{w}, \lambda_{T,i}; \mathbf{t}_i, \widetilde{\mathcal{B}}_{T,i}^-) = \frac{1}{|\widetilde{\mathcal{B}}_{T,i}^-|} \sum_{\mathbf{x} \in \widetilde{\mathcal{B}}_{T,i}^-} \exp(\text{sim}(E_I(\mathbf{x}), \mathbf{z}_{T,i})/\tau),$$

$$u_{I,i} \leftarrow \begin{cases} (1-\gamma)u_{I,i} + \gamma \widehat{g}(\mathbf{w}, \lambda_{I,i}; \mathbf{x}_i, \widetilde{\mathcal{B}}_{I,i}^-), & (\mathbf{x}_i, \mathbf{t}_i) \in \mathcal{B}, \\ u_{I,i}, & (\mathbf{x}_i, \mathbf{t}_i) \notin \mathcal{B}, \end{cases}$$

$$u_{T,i} \leftarrow \begin{cases} (1-\gamma)u_{T,i} + \gamma \widehat{g}(\mathbf{w}, \lambda_{T,i}; \mathbf{t}_i, \widetilde{\mathcal{B}}_{T,i}^-), & (\mathbf{x}_i, \mathbf{t}_i) \in \mathcal{B}, \\ u_{T,i}, & (\mathbf{x}_i, \mathbf{t}_i) \notin \mathcal{B}. \end{cases}$$

Finally, we can update the parameters $\mathbf{w}$ for image-text encoder networks by the stochastic gradient estimator.

$$\begin{aligned} \widehat{\nabla}_{\mathbf{w}} = \frac{1}{|\mathcal{B}|} \sum_{(\mathbf{x}_i, \mathbf{t}_i) \in \mathcal{B}} \Bigg[ & -2\nabla_{\mathbf{w}}\text{sim}(\mathbf{z}_{I,i}, \mathbf{z}_{T,i}) \\ & + \frac{\tau}{u_{I,i}} \nabla_1 \widehat{g}_I(\mathbf{w}, \lambda_{I,i}; \mathbf{x}_i, \widetilde{\mathcal{B}}_{I,i}^-) \\ & + \frac{\tau}{u_{T,i}} \nabla_1 \widehat{g}_T(\mathbf{w}, \lambda_{T,i}; \mathbf{t}_i, \widetilde{\mathcal{B}}_{T,i}^-) \Bigg]. \end{aligned}$$

## C. High-level Intuition for $\alpha$ Hyperparameter

The hyperparameter $\alpha$ allows GLOFND to adapt to different definitions of false negatives, which are inherently dependent on the desired level of granularity. For example, in a dataset like ImageNet, consider two classification settings: (1) a coarse-grained task of classifying between cars and animals, and (2) fine-grained task of classifying dog breeds. Two images of a dog from different breeds might be considered a false negative in the coarse-grained case (1), but not in the fine-grained one (2). Consequently, the optimal percentage of false negatives, controlled by $\alpha$, varies with the chosen granularity. By adjusting $\alpha$, GLOFND can flexibly align with different levels of semantic resolution, i.e., granularity.

The value of $\alpha$ can be set based on prior knowledge (e.g., expected rate of false negatives or desired granularity of learned representations) or tuned like other hyperparameters such as the temperature. If $\alpha$ is too low, GLOFND may fail to identify sufficient false negatives, leading to minimal impact on the learned representations, though not degrading performance, as setting $\alpha = 0$ is equivalent to disabling GLOFND. Conversely, if $\alpha$ is too high, GLOFND may identify too many false negatives. If these are filtered out during training, the reduced number of negative pairs can limit contrastive learning, potentially harming performance. For tuning, it is recommended to start with a low $\alpha$ and gradually increase it until no further performance gains are observed.

# D. More Details on Experiments

All the experiments are implemented using the PyTorch (Paszke et al., 2019) library. The unimodal experiments are run on a single NVIDIA A30 with 24GB memory size, while the bimodal experiments make use of a multi-node setup with 2 nodes, each with 2 NVIDIA A100 GPUs with 40GB each. The estimated amount of time to run a single experiment is 1.5 days for ImageNet100, and 14 hours for CC3M.

## D.1. Additional Details for Unimodal Experiments

**Experiment Setup.** Following prior work (Yuan et al., 2022), we pretrain a ResNet-50 (He et al., 2015) with a 2-layer $128 \times 128$ projection head on top of the backbone encoder. We use square root learning rate scaling ($0.075 \times \sqrt{\text{BatchSize}}$) with a cosine decay schedule without restart. Additionally, we apply a linear learning rate warm-up for 10 epochs, where the learning rate linearly increases to its maximum value.

We adopt the same augmentation pipeline as in SogCLR (Yuan et al., 2022), utilizing the torchvision implementation. This includes RandomResizedCrop (resizing to $224 \times 224$), random ColorJitter, RandomGrayscale, random GaussianBlur, RandomHorizontalFlip, and normalization using ImageNet statistics.

The network is pretrained for 200 epochs with a batch size of 128. We use the LARS optimizer (You et al., 2017) with a weight decay of $1e-6$ and momentum of 0.9. The temperature ($\tau$) is set to 0.1, and for SogCLR, $\gamma = 0.9$. For GLOFND, we set $\alpha = 0.01$ and tune the starting epoch from $\{70, 90, 110\}$, after which we begin updating $\lambda_i$ with Adam updates using a learning rate of 0.05, $\beta_1 = 0.9$, and $\beta_2 = 0.98$. For FNC, $\alpha = 0.01$ and the starting epoch is tuned from $\{10, 30, 50, 70, 90, 110, 130\}$, selecting the value that yields the best semi-supervised average performance.

**Linear Evaluation.** We evaluate GLOFND's ability to produce better representations through linear evaluation. Specifically, we freeze the encoder's weights at the last iteration of pretraining, remove its projection head, and train a linear classifier (a single fully connected layer) on top of the encoder's output.

Additionally, we employ a semi-supervised learning setup, using different fractions of labeled training data during linear evaluation. We train on random subsets of 100% (full dataset), 10%, 1%, and 0.1% of the training data. For each fraction, we report the top-1 accuracy on the validation set and average the performance across the different percentages to obtain the overall semi-supervised performance.

We train for 90, 285, 900, and 900 epochs corresponding to 100%, 10%, 1%, and 0.1% labeled data, respectively, with a batch size of 1024 and early stopping if the validation accuracy does not improve for 100 epochs. We use AdamW (Loshchilov & Hutter, 2019) with a weight decay of 0, momentum of 0.9, and a learning rate of 0.1. The same augmentation pipeline used in SogCLR is applied for linear evaluation. For training, we use RandomResizedCrop (resizing to $224 \times 224$), RandomHorizontalFlip, and normalization. For testing, we resize the images to $256 \times 256$, apply CenterCrop to $224 \times 224$, and normalize.

**Transfer Learning Datasets.** We additionally examine the transfer learning performance on Food-101 (Bossard et al., 2014), CIFAR-10 and CIFAR-100 (Krizhevsky, 2009), Stanford Cars (Krause et al., 2013), Describable Textures Dataset (DTD) (Cimpoi et al., 2014), Oxford-IIIT Pets (Parkhi et al., 2012), Caltech-101 (Li et al., 2022), and Oxford 102 Flowers (Nilsback & Zisserman, 2008). We follow the evaluation protocols in the papers introducing these datasets, i.e., we report top-1 accuracy for Food-101, CIFAR-10, CIFAR-100, Stanford Cars, and DTD; and mean per-class accuracy for Oxford-IIIT Pets, Caltech-101, and Oxford 102 Flowers. We report results on the test set and, for DTD, we report results only for the first split. Caltech-101 defines no train/test split, so we randomly select 20% of images per class to create the test set.

**Transfer Learning Evaluation.** We train an $\ell_2$-regularized multinomial logistic regression classifier on features extracted from the frozen pretrained network after removing the projector head. For each method, we select the pretrained network that achieved the highest semi-supervised average performance, as used in the semi-supervised results.

We employ L-BFGS and apply the same preprocessing as during validation in the linear evaluation setting: resizing to 256, center-cropping to 224, and normalizing. We report the best test performance across different $\ell_2$ regularization parameters, selecting from a range of 10 logarithmically spaced values between $10^{-6}$ and $10^5$.

## D.2. Additional Details for Bimodal Experiments

**Datasets.** For bimodal learning, we use the Conceptual Captions 3M (CC3M) (Sharma et al., 2018) dataset. Because some links have expired, our downloaded training set of CC3M contains $2,723,840$ image-text pairs. We evaluate the performance by leveraging the Datacomp Benchmark (Gadre et al., 2023), which includes 38 zero-shot downstream tasks. We report the average performance, named Datacomp. For each scenario, we select the model with the best Datacomp average and also report its average performance on two subsets of the tasks: zero-shot image classification on ImageNet and its different variants (IN & Variants), and zero-shot cross-modal image-text retrieval. IN & Variants includes ImageNet-1k (Russakovsky et al., 2015) and 6 ImageNet distribution shift datasets (i.e., ImageNet-Sketch (Wang et al., 2019), ImageNet-V2 (Recht et al., 2019), ImageNet-A (Hendrycks et al., 2021b), ImageNet-O (Hendrycks et al., 2021b), ImageNet-R (Hendrycks et al., 2021a), and ObjectNet (Barbu et al., 2019)). Retrieval tasks consist of Flickr30K (Plummer et al., 2017), MSCOCO (Lin et al., 2015), and WinoGAViL (Bitton et al., 2022).

**Experiment Setup.** Following previous work (Wei et al., 2024), we use a 12-layer transformer (Vaswani et al., 2017) as the text encoder and ResNet50 as the vision encoder. All experiments are conducted in a multi-node setting with 2 nodes, each equipped with two A100 40GB GPUs. We pretrain for 37 epochs with a global batch size of 1024. The AdamW (Loshchilov & Hutter, 2019) optimizer is used with $(\beta_1, \beta_2) = (0.9, 0.999)$, $\epsilon = 1e-8$, and a learning rate of $1e-3$. A weight decay of 0.1 is applied, with a warm-up period of 10k steps. The learning rate follows a cosine schedule, initially increasing linearly during the warm-up phase and then decreasing according to a cosine function. A cosine $\gamma$ schedule is employed, with a minimum $\gamma$ of 0.2 and decay epochs set to 18.

For SogCLR, the temperature parameter is set to 0.03. In FastCLIP, we set the initial temperature parameter to 0.07, $\rho$ to 6.5, and the learning rate for $\tau$ to $2e-4$. Additionally, the learning rate of $\tau$ decays to one-third of its original value when $\tau$ falls below 0.03. The complete set of hyperparameters is summarized in Table 4. We tune $\alpha \in \{5e-4, 1e-3\}$ and the starting epoch in $\{15, 20\}$. For SogCLR, we start using FNC/GLoFND after 15 epochs with $\alpha = 5e-4$, while for FastCLIP we start after 20 epochs with $\alpha = 1e-3$. For both losses, we initialize $\lambda_i = 1$ and use Adam updates with a learning rate of 0.05.

Table 4: Hyperparameters for FastCLIP Training

| Hyperparameter | CC3M |
|---|---|
| Optimizer | AdamW |
| $\beta_1, \beta_2$ | (0.9, 0.999) |
| $\epsilon$ | 1e-8 |
| Learning rate | 1e-3 |
| Weight decay | 0.1 |
| Warm-up steps | 10k |
| Cosine $\gamma$ min | 0.2 |
| Decay epochs | 18 |
| Temperature (SogCLR) | 0.03 |
| Initial temperature (FastCLIP) | 0.07 |
| $\rho$ (FastCLIP) | 6.5 |
| Learning rate of $\tau$ (FastCLIP) | 2e-4 |

## D.3. Additional Experimental Results

**Statistical Significance.**

We check for statistical significance between using the false negative approaches against not using them, which we consider the baseline. Thus, we compute p-values via a paired t-test between SogCLR + FNC/GLoFND and SogCLR baseline across multiple runs, with the alternative hypothesis testing for performance greater than the baseline.

We report in Tables 5 and 6 the p-values with respect to the baseline for the unimodal semi-supervised and transfer learning experiments respectively, and consider a standard significance level of 5%. For the semi-supervised scenario, we can observe GLoFND achieves a p-value below 0.018 on all scenarios, thus achieving statistically significant improvements in all scenarios. Moreover, GLoFND achieves statistical significance below the 1 % level on both the 100% and 1% scenarios.

For the transfer learning case, GLOFND achieves statistically significant improvements on 6 out of 8 datasets, while FNC only achieves it on 2 of 8.

Table 5: Unimodal semi-supervised linear evaluation p-values WRT the baseline (SogCLR). We color red those above the 0.05 threshold. The p-values are calculated via a paired t-test across multiple runs with the alternative hypothesis testing for performance greater than the baseline.

| Method | 100.0% | 10.0% | 1.0% | 0.1% |
|---|---|---|---|---|
| SogCLR + FNC | 0.011 | 0.035 | 0.014 | 0.024 |
| SogCLR + GLOFND | 0.002 | 0.011 | <0.001 | 0.018 |

Table 6: Unimodal transfer learning evaluation p-values WRT the baseline (SogCLR). We color red those above the 0.05 threshold. The p-values are calculated via a paired t-test across multiple runs with the alternative hypothesis testing for performance greater than the baseline.

| Method | CIFAR10 | CIFAR100 | Food101 | Caltech101 | Cars | DTD | Pets | Flowers |
|---|---|---|---|---|---|---|---|---|
| SogCLR + FNC | 0.158 | 0.056 | 0.219 | 0.249 | 0.235 | 0.080 | 0.032 | 0.036 |
| SogCLR + GLOFND | 0.143 | 0.005 | 0.021 | 0.024 | 0.006 | 0.123 | 0.002 | 0.005 |

**SimCLR Experiment**.

We evaluate the performance of GLOFND in conjunction with SimCLR. Unlike SogCLR, SimCLR identifies false negatives only within each mini-batch, necessitating the use of a larger batch size. Table 7 presents the linear evaluation results for SimCLR under the experimental setup described in Appendix D.1, using a batch size of 512.

Table 7: Linear evaluation results in unimodal semi-supervised scenario. We train the linear classifiers with different percentages of randomly sampled labeled training data and present their top-1 accuracies (%) on the validation set. We include the overall average and, in parentheses, its improvement WRT SimCLR baseline.

| Method | 100.0% | 10.0% | 1.0% | 0.1% | Average |
|---|---|---|---|---|---|
| SimCLR | 76.88 | 73.38 | 66.40 | 33.56 | 62.56 |
| + FNC | 76.90 | 73.10 | 64.88 | 34.20 | 62.27 |
| + GloFND | **77.14** | **73.66** | **66.50** | **35.58** | **63.22** |

**CLIP fine-tuning**.

To evaluate the effectiveness of GLOFND in fine-tuning large pretrained models to mitigate the impact of false negatives, we fine-tune OpenAI's ResNet-50 CLIP model on CC3M. The experimental setup closely follows Appendix D.2, with the key difference being that we initialize from OpenAI's pretrained weights and train for 15 epochs, using FNC/GLOFND after the first epoch ($\alpha = 10^{-4}$). Validation image-text retrieval results are reported in Table 8.

Table 8: We fine-tune OpenAI's ResNet-50 CLIP model on CC3M and report image-text retrieval results on its validation set.

| Method | IR@1 | IR@5 | IR@10 | IR Average | TR@1 | TR@5 | TR@10 | TR Average |
|---|---|---|---|---|---|---|---|---|
| Base | 27.58 | 48.17 | 57.53 | 44.43 | 26.30 | 48.14 | 57.69 | 44.04 |
| GloFND | 36.07 | 58.94 | 67.22 | 54.08 | **35.76** | 59.19 | 67.44 | 54.13 |
| + FNC | 33.69 | 58.01 | 67.51 | 53.07 | 33.61 | 57.95 | 67.53 | 53.03 |
| + GloFND | **36.52** | **59.44** | **68.02** | **54.66** | 35.71 | **59.27** | **67.97** | **54.32** |

**More examples of false negatives identified by GLOFND**.

Figure 7 shows examples of false negatives identified by GLOFND for ImageNet100 with $\alpha = 0.01$ during training. We can observe that the number of false negatives identified is not constant for all anchors since we are using a dynamic threshold for each anchor, as opposed to a mini-batch top k approach. Moreover, the false negatives identified by GLOFND are

Anchor     False negatives     Anchor     False negatives

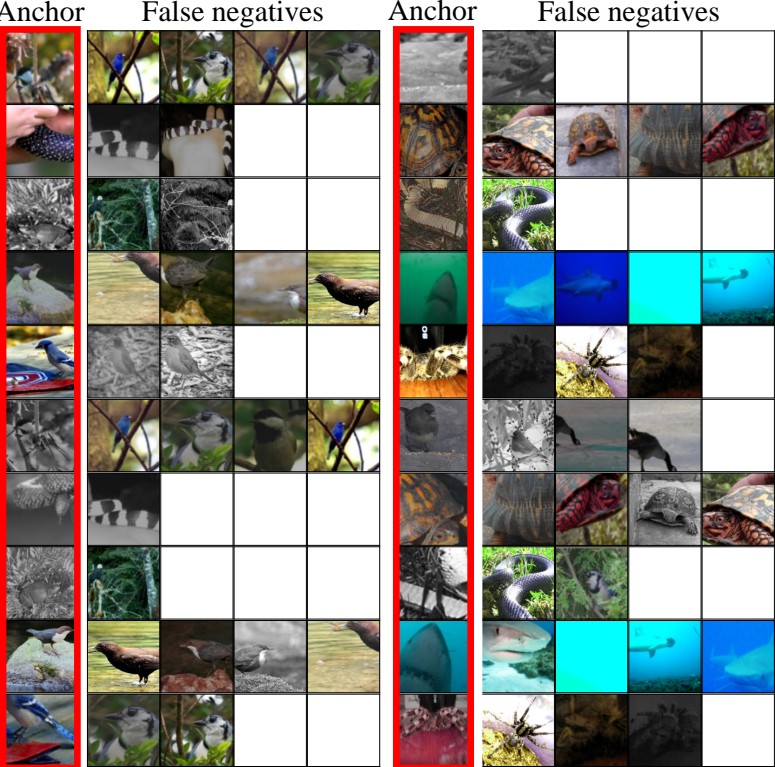

Figure 7: Examples of false negatives identified for ImageNet100 by GLOFND with $\alpha = 1\%$. The left column is the anchor image and the rest are identified false negative samples.

semantically similar to the anchor image, which is what we aim to achieve.

