# OpenReview forum: "Discovering Global False Negatives On the Fly for Self-supervised Contrastive Learning"
_ICML.cc/2025/Conference — ICML 2025 poster_

### Official Review · Reviewer_KEsn · 2025-03-10

**Overall Recommendation:** 4

**Summary:**

This paper presents an approach to detect false negative pairs in contrastive learning. Contrary to previous approaches, they work by detecting FN globally on the pretraining dataset, and they are computationally efficient as they apply SGD for each anchor and do not require to compute clustering for the whole dataset. They conduct extensive experiments to compare and validate the approach.

**Claims And Evidence:**

This paper claims to have a efficient and effective approach for FN detection in contrastive learning. They do not overclaim, and they conduct the necessary experiments to prove their approach works well.

It would have been nice to see other downstream tasks than classification to validate further the quality of the learned representation. The contributions should state clearly that the findings are only validated in the context of classification and that further experiments are needed to evaluate performance of representation learning for other tasks.

**Essential References Not Discussed:**

None

**Experimental Designs Or Analyses:**

Experiments are sufficient to support the claim about the FN detection method being good and efficient.

It would be good to display computational overhead results compared with other approaches in the core of the paper, since it is part of the claimed advantages of the method.

**Methods And Evaluation Criteria:**

Proposed datasets and metrics are sufficient and appropriate for evaluation of the method.

**Other Comments Or Suggestions:**

See above

**Other Strengths And Weaknesses:**

It would be good to provide the reader with a rough idea of the scale of the amount of FN in practical CL scenarios. For example, for practical CL training on ImageNet, how many FN do we have on average? Also, it would be good to give an idea of the impact these FN have on final results (results on a particular downstream tasks can be shown in an early figure). This should be mentioned explicitely in the introduction, as it would help grasping the importance of the proposed line of work.

By removing examples that are close to each other in the latent space as FN, isn't there a risk of removing hard true negatives, thus making the contrastive learning too simple and thus reducing the performance of the representation learned? --> In other words, could it decrease the performance for fine grained classification downstream tasks? This could be tested in the experiments by including fine grained datasets.

"Moreover, we assume that the top α% most similar negative data share similar semantics with the anchor data based on their current representations" --> This is a rather strong assumption. This should be backed up by experiments using the labels. What proportions of detected FN are actually FN in practice?

**Questions For Authors:**

See above

**Relation To Broader Scientific Literature:**

The literature review is very complete, it flows logically and presents a nice introduction to the topic.

**Theoretical Claims:**

I did not check the proof in the supplementary material in full details, but it seems correct.

---

> ### Author Rebuttal · Authors · 2025-04-01
>
> Thanks for your review and questions.
>
> **Q1: It would have been nice to see other downstream tasks than classification to validate further the quality of the learned representation.**
>
> **A:** Thanks for pointing this out. We agree that the unimodal performance has only been validated in the context of classification. In the bimodal scenario, please note we use the Datacomp Benchmark. This benchmark includes 38 zero-shot downstream tasks, which also include cross-modal image-text retrieval tasks. As suggested, we will modify the contributions to clarify the tasks used for validation.
>
>
> **Q2: Display computational overhead results in the core of the paper.**
>
> **A:** Thanks for your comment. We will move the computational overhead result to the main paper.
>
>
> **Q3: Introduction, for practical CL training on ImageNet, how many FN do we have on average? Also, it would be good to give an idea of the impact these FN have on final results (results on a particular downstream tasks can be shown in an early figure).**
>
> **A:** During training on ImageNet100, the empirical average of FN is 1%, that's around 20k FN per batch with a batch size of 1024 and 325 with a batch size of 128. Thanks for your suggestion. We agree both of these questions would provide a better idea of the importance of this line of work. We will make changes to the introduction to address both.
>
>
> **Q4: GloFND performance for fine-grained classification downstream tasks?**
>
> **A:** Thanks for the question, this is something we could have done a better job pointing out. Table 2 presents the performance on several fine-grained downstream datasets such as Stanford Cars, Oxford 102 Flowers, Oxford-IIIT Pets, Caltech-101, and Food-101.
>
>
> **Q5: About the top $\alpha$% assumption. What proportions of detected FN are actually FN in practice?**
>
> **A:** Table 1 presents "False Negative Identification" metrics, which evaluate what proportion of detected FN are actually FN in practice. As it can be observed, GloFND achieves much better precision, recall, and f1-score compared to FNC. Answering your question, the last epoch mean for GloFND is 48.40% of predicted FN are true FN, compared to 27.57% for FNC.

---

### Official Review · Reviewer_UmKd · 2025-03-11

**Overall Recommendation:** 2

**Summary:**

Previous contrastive learning methods may generate negative sample pairs with similar semantics when constructing negative samples. Different from them, this paper introduces a method that automatically learns on the fly the threshold for each anchor data to identify its false negatives during training. Meanwhile, it can globally detect false negative samples rather than locally within the mini-batch. Experiments are done to verify the effectiveness of the method.

**Claims And Evidence:**

The claims made in the opinion are supported by clear and compelling evidence and further validated by experiments.

**Essential References Not Discussed:**

No.

**Experimental Designs Or Analyses:**

This paper has basically no problems in terms of experimental design and analysis, but there is one problem is that the authors claim that GLOFND is not limited to any specific contrastive learning (CL) technique, however, there does not seem to be direct evidence supporting this claim in the experiments.

**Methods And Evaluation Criteria:**

Yes, the method proposed in this paper is a good research direction in the field of contrastive learning.

**Other Comments Or Suggestions:**

(1)The authors mention that previous methods require computing cosine similarity across the entire dataset when selecting the most similar negative samples to the anchor. However, it appears that they have not effectively addressed this computational overhead.
(2)The authors should apply their method to a broader range of contrastive learning (CL) approaches to further validate its effectiveness and strengthen its credibility.

**Other Strengths And Weaknesses:**

Strengths：
(1)In this paper, a novel method is proposed to automatically find false negative samples, and its view is proved by experiments. The article has clear ideas. The method is simple and effective.
(2)The structure of this paper is clear.
(3)The proposed method has better performance and experimental results in some cases
Weaknesses:
(1)The authors claim that GLOFND is not limited to any specific contrastive learning (CL) technique. However, they only integrate it with SogCLR and do not demonstrate its effectiveness on classical contrastive learning algorithms, thereby reducing the credibility of their claims.
(2)The novelty of this method is insufficient, and its generalizability does not seem to be well demonstrated in the paper.
(3)The experiments in this paper seem to be insufficient. The author focuses on how to find false negative samples, but ignores the generality research in specific comparative learning or self-supervised learning methods.
(4)The author's focus does not seem to be on downstream task performance, and adding some downstream task evaluations, such as object detection, might be more convincing.

**Questions For Authors:**

There's no question.

**Relation To Broader Scientific Literature:**

In this paper, the method "GLOFND" is proposed from the perspective of detecting false negative samples. However, how to reasonably define false negative samples and eliminate them is an important research direction of contrast learning, which has a certain impact on the improvement of model performance.

**Theoretical Claims:**

The false negative sample detection method proposed in this paper has no obvious problems in theory.

---

> ### Author Rebuttal · Authors · 2025-04-01
>
> Thanks for your review and questions.
>
> **Q1: About GloFND limitations to a specific contrastive learning (CL) technique and integration on classical contrastive learning algorithms.**
>
> **A:** Thanks for this question, this is something we could have done a better job explaining. To clarify, the computation of $\lambda$ in GloFND is independent of the contrastive loss used, as it relies solely on the embedding similarity of negative pairs. This makes it applicable across different contrastive learning methods. In our experiments, we applied GloFND to unimodal SogCLR, bimodal SogCLR and FastCLIP. As you suggested, we have additionally run unimodal GloFND with SimCLR with results shown in Reviewer tdJR's Q2 answer. However, it is worth noting that prior work [1] has shown that SogCLR outperforms SimCLR. Additionally, SimCLR requires a large batch size, and its performance can be more sensitive to the impact of false negatives as batch size increases.
>
> [1] Yuan, Zhuoning, Yuexin Wu, Zi-Hao Qiu, Xianzhi Du, Lijun Zhang, Denny Zhou, and Tianbao Yang. “Provable Stochastic Optimization for Global Contrastive Learning: Small Batch Does Not Harm Performance.” arXiv, September 20, 2022. http://arxiv.org/abs/2202.12387.
>
>
> **Q2: The novelty of this method is insufficient, and its generalizability does not seem to be well demonstrated in the paper.**
>
> **A:**  We appreciate the reviewer’s feedback and would like to clarify the novelty and generalizability of our approach. Formulating false negative discovery as identifying the k-th largest similarity across the **entire dataset** is, to the best of our knowledge, a novel contribution. This formulation enables a simple yet effective algorithm based on stochastic optimization for efficient computation.
>
> We have conducted experiments to demonstrate its applicability in unimodal CL (semi-supervised and transfer learning) and bimodal CL. Moreover, to further demonstrate generalizability, we have conducted additional experiments with SimCLR, reinforcing its broad applicability (see the previous question). We appreciate the reviewer’s concerns and are open to suggestions that could further strengthen this aspect.
>
>
> **Q3: The experiments in this paper seem to be insufficient. The author focuses on how to find false negative samples, but ignores the generality research in specific comparative learning or self-supervised learning methods.**
>
> **A:** To address your concern, we have conducted additional experiments using SimCLR (see Reviewer tdJR Q2). We have also conducted an experiment fine-tuning OpenAI's CLIP model on CC3M (see Reviewer tdJR Q3).
>
>
> **Q4: The author's focus does not seem to be on downstream task performance.**
>
> **A:** We would like to draw the reviewer’s attention to Tables 2 and 3, which present results on downstream task performance. In Table 2, we pretrain on ImageNet100, after which a logistic regression classifier is trained on top of the frozen embeddings for multiple unimodal downstream datasets. In Table 3, we pretrain on CC3M and evaluate performance on 38 zero-shot downstream tasks using the DataComp benchmark. Notably, GloFND improves downstream performance in most scenarios.
>
> **Q5: Addressing computational overhead of computing cosine similarity across the entire dataset when selecting the most similar negative samples to the anchor.**
>
> **A:** Thank you for the question. While GloFND computes a global threshold for the entire dataset, it does not require computing cosine similarity across all data points in the dataset. Instead, all computation is done in the mini-batch. This is what makes GloFND shine.
>
> GloFND frames the problem as a stochastic optimization task, allowing for efficient computation. Equation 4 details how the $\lambda_I$ values are obtained, and as shown, GloFND relies only on mini-batch computations to optimize $\lambda_i$.
>
> Importantly, the pairwise similarities are already computed as part of the contrastive loss, meaning the only additional computational overhead introduced by GloFND is:
> 1. Updating the $\lambda$ values for samples in the mini-batch, which involves a simple gradient computation (Equation 4).
> 2. Filtering false negatives, which is also done via matrix operations by comparing similarities against the computed $\lambda$ values and applying masking.
>
> Both can run efficiently on GPUs. Overall, our method consists of basic matrix computations and runs in linear time with respect to the number of pairs in a batch $O(B^2)$, where $B$ is the batch size). This overhead is minimal compared to the cost of cosine similarity computations and the forward/backward passes. For the unimodal case, the per-epoch computation time increases by only 2% (from 427s to 435s), demonstrating the efficiency of our approach.

---

### Official Review · Reviewer_tdJR · 2025-03-13

**Overall Recommendation:** 3

**Summary:**

In this work, authors propose GLOFND, a way to find and automatically threshold false negative samples during self-supervised training with contrastive learning. The proposed method works by determining adaptive thresholds $\lambda_i$ for each anchor $i$ which, thanks to the optimization-based approach, are global to the entire dataset and not limited to the current minibatch. Contrarily to other existing methods which require to compute similarity across all possible pairs, GLOFND does not introduce significant complexity in the training.

Authors empirically test their approach on standard vision benchmarks such as ImageNet-100, CIFAR, DTD Caltech and Oxford datasets.

## Update after rebuttal

After reading the author's comments and other reviewers' comments, I maintain my score.

**Claims And Evidence:**

- This work identifies the issue of determining false positive pairs in a global and efficient manner

- The proposed approach is in line with the authors' goal

- The proposed approach can be integrated into to many existing contrastive learning framework

**Essential References Not Discussed:**

N/A

**Experimental Designs Or Analyses:**

I think that the main weakness of this work lies in the experimental setting:

- Larger datasets such as ImageNet-1k are missing (I think experiments on it should be doable)

- The only comparison is with FNC, which selects local false positive samples in a minibatch

- GLOFND is only applied to SogCLR

For these reasons i think that:

- Some more experiments are required, showcasing GLOFND applied on other baselines (SimCLR, VICReg, Barlow Twins, etc.)

- GLOFND requires the network to be sufficiently trained to work: for me this makes sense, and I think that applying GLOFND for fine-tuning large pre-trained model would be for an excellent use case. I would like to see some experiments in fine-tuning some existing large models to reduce the impact of false-positive pairs (authors only tested with SogCLR and FastCLIP)

- A comparison with finding global thresholds in the entire dataset (even if computationally expensive) should be added

**Methods And Evaluation Criteria:**

- The evaluation criteria ideally are okay, my main concern is in the lack of comparison and well-established baselines [see experimental design or analyses].

- The ablation study is thorough

**Other Comments Or Suggestions:**

- L190-L191 are not clear

**Other Strengths And Weaknesses:**

- The paper is very well written and the objective is clear

- I think Eq. 2 would benefit from a clearer explanation

**Questions For Authors:**

- In Fig. 5d, authors show the results of GLOFND with varying starting epochs; I wonder why the performance decreases after epoch 70?

- Improvements in the bimodal setting are less significant than in the unimodal setting, what could be the reason for it?

**Relation To Broader Scientific Literature:**

This work proposes an adaptive method to find thresholds for filtering out false positive samples. It may represent an interesting contribution to the field.

**Theoretical Claims:**

- I think the theoretical claims are well motivated, although I would make a stronger connection to meta-learning literature (as essentially the thresholds $\lambda_i$ will change the main objective function)

---

> ### Author Rebuttal · Authors · 2025-04-01
>
> Thanks for your review and questions.
>
> **Q1: Experiment on larger dataset than ImageNet**
>
> **A:** We would like to point the reviewer to Table 3, where we tested GloFND on CC3M, which is larger than ImageNet-1k, with 2.7 million image-text pairs.
>
>
> **Q2: GLOFND applied on other baselines (SimCLR, VICReg, Barlow Twins, etc.)**
>
> **A:** We agree that it would be interesting to apply GloFND to other contrastive losses. To the best of our knowledge, VICReg and Barlow Twins do not use negative pairs, so the issue of false negative detection is not directly applicable. Regarding SimCLR, SogCLR has been previously shown [1] to outperform SimCLR, which requires a large batch size. Nevertheless, we have tried SimCLR using a batch size 512 with results shown below.
>
> | Method   | 100%  | 10%   | 1%    | 0.1%  | Average |
> | -------- | ----- | ----- | ----- | ----- | ------- |
> | Baseline | 76.88 | 73.38 | 66.40 | 33.56 | 62.56   |
> | FNC      | 76.90 | 73.10 | 64.88 | 34.20 | 62.27   |
> | GloFND   | **77.14** | **73.66** | **66.50** | **35.58** | **63.22**   |
>
> [1] Yuan, Zhuoning, Yuexin Wu, Zi-Hao Qiu, Xianzhi Du, Lijun Zhang, Denny Zhou, and Tianbao Yang. “Provable Stochastic Optimization for Global Contrastive Learning: Small Batch Does Not Harm Performance.” arXiv, September 20, 2022. http://arxiv.org/abs/2202.12387.
>
>
> **Q3: Fine-tuning large pre-trained model**
>
> **A:** We have fine-tuned OpenAI's CLIP model on CC3M using GloFND. The results on CC3M's validation set are shown below.
>
> | Method   | IR@1    | IR@5      | IR@10        | IR Avg | TR@1       | TR@5            | TR@10         | TR Avg |
> | -------- | ------- | --------- | ------------ | ------ | ---------- | --------------- | ------------- | ------ |
> | Baseline | 36.07   | 58.94     | 67.22        | 54.08  | **35.76**      | 59.19           | 67.44         | 54.13  |
> | FNC      | 33.69 | 58.01 | 67.51 |   53.07     | 33.61 | 57.95 | 67.53 |      53.03  |
> | GloFND   | **36.52**   | **59.44**     | **68.02**        | **54.66**  | 35.71      | **59.27**           | **67.97**         | **54.32**  |
>
>
>
> **Q4: Comparison with finding global thresholds in the entire dataset**
>
> **A:** We agree such a comparison would be beneficial. Kindly note that computing the global threshold in the entire dataset would need to be done every iteration, which is intractable. Instead, we freeze the encoder network and compare GloFND with the (estimated) global thresholds. You can find this analysis in Section 4.3 (iii).
>
> **Q5: Why does the performance decrease after epoch 70?**
>
> **A:** Thanks for this question. We hypothesize the reason is that the total number of training epochs is fixed to 200. That is, there is a trade-off between starting at a later epoch (and having a better "sufficiently" trained network), and the amount of training epochs for GloFND to positively impact. Starting at a later epoch entails less training time with GloFND which reduces the potential improvement of removing false negatives. FNC shows a similar pattern, with its performance dropping after 110.
>
> **Q6: Improvements in the bimodal setting are less significant than in the unimodal setting**
>
> **A:** Thanks for pointing this out. We agree with your observation. We hypothesize that this is due to the training time because the bimodal dataset is much larger.  While we train GloFND for 130 epochs in the unimodal setting, training for only 22 epochs in the bimodal setting makes the value of $\lambda$ less stable.

---

### Decision · Program_Chairs · 2025-05-01

**Decision:**

Accept (poster)

**Comment:**

The paper proposes a method for learning per anchor similarity thresholds that separate false negatives from true negatives during contrastive learning.  Reviewer opinion is split, with tdJR and KEsn recommending accept, while UmKd recommends reject.  The rebuttal resolves some, but not all reviewer concerns.

Reviewer UmKd notes that "novelty of this method is insufficient, and its generalizability does not seem to be well demonstrated" and asks for experiments on "downstream task evaluations, such as object detection".  The rebuttal does not fully resolve this concern as it points to Tables 2 and 3, which cover only classification and retrieval tasks, rather than tasks such as object detection or segmentation that require more detailed output.

Reviewer KEsn raises concern over a strong assumption made in the paper about the structure of the data: "we assume that the top $\alpha$% most similar negative data share similar semantics with the anchor data based on their current representations."  The rebuttal responds with clarifications about $\alpha$, but does not address the core of this concern.  Specifically, $\alpha$ is a hyper-parameter; if chosen correctly it can yield good results (as demonstrated in experiments), but the paper chooses $\alpha$ in an ad-hoc, heuristic manner.

Overall, limitations in novelty and generality are balanced against experiments that demonstrate the efficiency of the approach and improved results.